# Serotype distribution and antimicrobial resistance of human *Salmonella enterica* in Bangui, Central African Republic, from 2004 to 2013

**Sebastien Breurec**[1,2,3,4]*, **Yann Reynaud**[2], **Thierry Frank**[1], **Alain Farra**[1],
**Geoffrey Costilhes**[5], **François-Xavier Weill**[5], **Simon Le Hello**[5]

**1** Laboratoire de Bactériologie, Institut Pasteur, Bangui, Central African Republic, **2** Unité Transmission, Réservoir et Diversité des Pathogènes, Institut Pasteur de Guadeloupe, Les Abymes, France, **3** Faculté de Médecine Hyacinthe Bastaraud, Université des Antilles, Pointe-à-Pitre, France, **4** Laboratoire de Microbiologie clinique et environnementale, Centre Hospitalier Universitaire de Pointe-à-Pitre/les Abymes, Pointe-à-Pitre, France, **5** Unité des Bactéries Pathogènes Entériques, Centre National de Référence des *Escherichia coli*, *Shigella* et *Salmonella*, World Health Organization Collaborative Centre for typing and antibiotic resistance of *Salmonella*, Institut Pasteur, Paris, France

* sbreurec@gmail.com

**Data Availability Statement:** All relevant data are within the manuscript and its Supporting Information files.

## Abstract

### Background

Limited epidemiological and antimicrobial resistance data are available on *Salmonella enterica* from sub-Saharan Africa. We determine the prevalence of resistance to antibiotics in isolates in the Central African Republic (CAR) between 2004 and 2013 and the genetic basis for resistance to third-generation cephalosporin (C3G).

### Methodology/Principal findings

A total of 582 non-duplicate human clinical isolates were collected. The most common serotype was Typhimurium (n = 180, 31% of the isolates). A randomly selected subset of *S.* Typhimurium isolates were subtyped by clustered regularly interspaced short palindromic repeat polymorphism (CRISPOL) typing. All but one invasive isolate tested (66/68, 96%) were associated with sequence type 313. Overall, the rates of resistance were high to traditional first-line drugs (18–40%) but low to many other antimicrobials, including fluoroquinolones (one resistant isolate) and C3G (only one ESBL-producing isolate). The extended-spectrum beta-lactamase (ESBL)-producing isolate and three additional ESBL isolates from West Africa were studied by whole genome sequencing. The $bla_{CTX-M-15}$ gene and the majority of antimicrobial resistance genes found in the ESBL isolate were present in a large conjugative IncHI2 plasmid highly similar (> 99% nucleotide identity) to ESBL-carrying plasmids found in Kenya (*S.* Typhimurium ST313) and also in West Africa (serotypes Grumpensis, Havana, Telelkebir and Typhimurium).

**Funding:** The French National Reference Centre for E. coli, Shigella, and Salmonella is funded by the Institut Pasteur and Santé Publique France. The Unit for Pathogenic Enteric Bacteria is part of the Integrative Biology of Emerging Infectious Diseases Laboratory of Excellence funded by the French Government "Investment in the Future" programme (Grant no. ANR-10-LABX-62-IBEID, https://anr.fr/en/investments-for-the-future/investments-for-the-future/). This project was supported by the WHO Laboratory twinning initiative "Improvement of diagnostic and surveillance capacity for Salmonella in the Central African Republic" (WHO reference grant 2011/178985-0). The funders had no role in the study design, data collection and analysis, decision to publish, or preparation of the manuscript.

**Competing interests:** The authors have declared that no competing interests exist.

## Conclusions/Significance

Although the prevalence of ESBL-producing *Salmonella* isolates was low in CAR, we found that a single IncHI2 plasmid-carrying $bla_{CTX-M-15}$ was widespread among *Salmonella* serotypes from sub-Saharan Africa, which is of concern.

## Author summary

*Salmonella enterica* infections are common causes of bloodstream infection in sub-Saharan Africa and associated with a high mortality rate. Levels of multidrug resistance have become alarmingly high. Then, third-generation cephalosporin (C3G) and fluoroquinolones have become standard for first-line empirical treatment. Recently, C3G-resistant *Salmonella* populations have emerged and spread over all continents. This resistance is mainly mediated by acquired extended-spectrum beta-lactamase (ESBL) genes carried by mobile genetic elements such as plasmids. We report here the prevalence of resistance to antibiotics in isolates in the Central African Republic (CAR) between 2004 and 2013 and the genetic basis for resistance to C3G. Overall, resistance rates to antimicrobials were low during the study period, for all classes other than conventional antimicrobials, confirming recommendations for first-line treatment based on C3G and fluoroquinolones. Only one ESBL-producing isolate was recovered. The ESBL gene and the majority of antimicrobial resistance genes found were present in a large plasmid highly similar to ESBL-carrying plasmids found in East and West Africa, highlighting its significant role in the spread of ESBL genes in *Salmonella* isolates in sub-Saharan Africa. These finding have implications for treatment of salmonellosis and support the growing necessity for increased microbiological surveillance based on networks of clinical laboratories in order to control dissemination of antibiotic resistance among *Salmonella* isolates.

## Introduction

*Salmonella enterica* serotypes Typhi, Paratyphi A, Paratyphi B d-tartrate negative and Paratyphi C are grouped as typhoidal *Salmonella*, and other serotypes are described as non-typhoidal *Salmonella* (NTS). Human *Salmonella* infections are generally either typhoid and paratyphoid fever, systemic diseases caused by typhoidal *Salmonella*, or gastroenteritis caused by a large number of NTS serotypes. Although most cases of salmonellosis due to NTS are self-limiting, they have emerged as a prominent cause of life-threatening bloodstream infections in sub-Saharan Africa, with approximately 388 000 deaths in 2010 [1].

*S. enterica* serotype Typhimurium (referred to here as *S*. Typhimurium) sequence type (ST) 313 has been described as the primary cause of invasive salmonellosis in sub-Saharan Africa, with mortality rates > 25%. Whole-genome analysis revealed that these ST313 strains belong to two dominant genetic lineages, I and II, which emerged independently within the past 40–50 years and spread across sub-Saharan Africa, in close temporal association with the current HIV pandemic [2]. Clonal replacement of lineage I by lineage II was observed in the mid-2000s, perhaps due to acquisition of chloramphenicol resistance [2].

Currently, there is no commercially available anti-*Salmonella* vaccine for controlling invasive NTS. Prompt, effective management with antimicrobials remains the main option for management of the disease; however, antimicrobial resistance emerged several decades ago, initially to the traditional first-line drugs such as chloramphenicol, ampicillin and

cotrimoxazole. Third-generation cephalosporins (C3G) and fluoroquinolones have since become standard first-line empirical treatment [2]. Recently, C3G-resistant (C3GR) *Salmonella* populations have emerged and spread in all continents, including Africa [3].

The Central African Republic (CAR) is a resource-limited country in equatorial Africa (ranked 188/189 on the Human Development Index in 2018). In a case–control study conducted in Bangui, the capital, between 2011 and 2013 among children under 5 years of age, *S. enterica* was isolated in 4% of children presenting with diarrhoeal illness; rotavirus and *Cryptosporidium hominis/parvum* were the two pathogens most frequently recovered [4]. The study's primary objective was to report on the prevalence of resistance to antibiotics in *S. enterica* strains isolated from patients in CAR between 2004 and 2013 to determine whether recommendations for first-line empirical treatment based on C3G and fluoroquinolones are relevant. Secondary objectives were i) to investigate the genetic background of serotype Typhimurium to determine whether ST313 is the primary cause of invasive salmonellosis, iii) to determine the genetic basis for resistance to C3G using whole genome sequencing (WGS).

## Material and methods

### Ethical approval

This retrospective study is based on a retrospective data collection issued from the routine bacteriological diagnostic activity within the private clinical laboratory located at Institut Pasteur in Bangui. Data consisted of anonymised laboratory results devoid of individual patient information or identifiers. Ehical approval and individual informed consent were not necessary. We submitted the study protocol to the WHO representative in Bangui, the Institut Pasteur in Bangui and the Ministry of Public Health in CAR which gave their approval.

### Bacterial strains, serotyping, susceptibility, transferability of extended-spectrum beta-lactamase

Our private clinical laboratory, the major one in the capital, performed biological analysis on specimens from its patients but also on samples from other hospitals and clinical laboratories. All non-duplicate *S. enterica* clinical isolates collected between January 2004 and December 2013 were included, encompassing those previously described [4]. If more than one isolate with the same serotype and antimicrobial resistance phenotype was recovered from the same patient, only the first was retained for the analysis. Date of isolation and the nature of the biological sample were recorded.

The strains were serotyped on the basis of somatic O and phase 1 and phase 2 flagellar antigens by agglutination tests with antisera (Bio-Rad, Marnes-La-Coquette, France), as specified in the White-Kauffmann-Le Minor scheme (preliminary serotyping at the Institut Pasteur in Bangui, complete serotyping at the Institut Pasteur in Paris). Susceptibility to amoxicillin (20 μg), amoxicillin-clavulanic acid (20–10 μg), ticarcillin (75 μg), cefalotin (30 μg), cefoxitin (30 μg), cefotaxime (5 μg), ceftazidime (10 μg), imipenem (10 μg), amikacin (30 μg), gentamicin (10 μg), spectinomycin (10 μg), streptomycin (10 UI), nalidixic acid (30 μg), ciprofloxacin (5 μg), chloramphenicol (30 μg), cotrimoxazole (1.25–23.75 μg) and tetracycline (30 μg) was tested by the disc diffusion method on Mueller-Hinton agar (Bio-Rad), and production of ESBL was detected by the double-disc synergy (DDS) test. Minimum inhibitory concentrations (MICs) were determined with E-test strips (BioMerieux, Marcy L'Etoile, France). Isolates were classified as resistant, intermediate or susceptible according to the 2018 guidelines of CA-SFM/EUCAST (http://www.sfm-microbiologie.org). Isolates of intermediate susceptibility were not classified with the resistant ones for the analysis of the data.

A conjugation experiment was performed as described previously [5].

### Clustered regularly interspaced short palindromic repeats polymorphism (CRISPOL) typing and multilocus sequence typing (MLST) of *S.* Typhimurium isolates

Total DNA was extracted with the Instagene kit (Bio-Rad, Marnes-la-Coquette, France). CRIS-POL typing and MLST was performed on the Luminex platform (Luminex Coporation, Austin, TX, USA) at the Institut Pasteur in Paris, as previously described [6].

### Whole-genome sequencing

WGS was performed on the ESBL-producing *S.* Typhimurium isolate S1027072 identified in this study and on three *Salmonella* isolates (serotypes Grumpensis, Havana and Telelkebir) recovered from West Africa in 2007 and 2008 and previously described, at the Institut Pasteur in Paris [3] (Table 1). These three comparison isolates produced a CTX-M-15 ESBL carried on a large IncHI2 plasmid.

DNA extraction was performed with the MagNAPure 96 system (Roche). The libraries were prepared with a Nextera XT kit (Illumina) and sequencing with the NextSeq 500 system (Illumina), generating 150-bp paired-end reads, yielding a mean 105-fold coverage. Reads were trimmed and filtered with AlienTrimmer [7] and a quality Phred score threshold of 28 on a minimum length of 70 nucleotides.

### Plasmid assemblies and syntenic analysis

In order to reconstruct plasmids in ESBL-producing *S.* Typhimurium ST313 isolate S1027072, reads were mapped with Bowtie 2 software [8] against two assembled, annotated plasmids found in an *S.* Typhimurium ST313 isolate from Kenya [9]: pKST313, an IncHI2 plasmid encoding ESBL CTX-M-15 and pSBLT, an IncFII plasmid similar to the previously described pSLT-BT found in strain D23580 isolated in Malawi in 2004 [2]. Selected mapped reads were

**Table 1. Plasmids used for syntenic analysis versus pKST313.**

| Strain | Species/Serovar | Country | Origin | Year | Plasmid | Accession number | Size (bases) | Percentage identity vs pKST313 |
|---|---|---|---|---|---|---|---|---|
| KST313 | *S.*[a] Typhimurium | Kenya | Human | 2009–2012 | pKST313 | LN794248 | 300 375 | – |
| S1027072 | *S.* Typhimurium | CAR | Human | 2008 | pCARST313 | PRJNA540305 | 268606[d] | 99.98 |
| 08–3663 | *S.* Grumpensis | Senegal | Human | 2008 | p08-3663 | PRJNA540305 | 280987[d] | 99.98 |
| 07–0319 | *S.* Havana | Mali | Human | 2007 | p07-0319 | PRJNA540305 | 282330[d] | 99.98 |
| 07–1331 | *S.* Telelkebir | Mali | Human | 2007 | p07-1331 | PRJNA540305 | 282370[d] | 99.99 |
| A54560 | *S.* Typhimurium | Malawi | Human | 2009 | pSTm-A54650 | NC_024983.1 | 309 406 | 99.76 |
| EB-247 | *E.*[b] *bugandensis* | United Republic of Tanzania | Human | 2010 | pEB247 | LN830952.1 | 298 984 | 99.80 |
| CRENT-193 | *Enterobacter* sp. | Republic of Korea | Human | 2013 | pCRENT-193_1 | NZ_CP024813.1 | 298 989 | 99.82 |
| Ec21617 | *E.*[c] *coli* | Taiwan | Human | 2013 | pEc21617-310 | MG878867.1 | 309 608 | 99.67 |

[a]*Salmonella*

[b]*Enterobacte*r

[c]*Escherichia*

[d]Not fully assembled

assembled with SPAdes [10], and the quality of the assembly was checked with QUAST software [11]. Plasmidic contigs were ordered against the reference plasmids pKST313 and pSBLT with ABACAS software to produce the putative plasmids pCARST313 and pCAR_SBLT, respectively [12]. The contigs were further annotated with Prokka [13]. Comparison files between putative plasmids and their reference were generated with BRIG software [14].

To better describe worldwide circulation of $bla_{CTX-M-15}$-carrying IncHI2 plasmids, we used the same approach to compare pCARST313 with IncHI2 plasmids found in three *Salmonella* isolates from West Africa (Table 1). In addition, the nucleotide sequence of pCARST313 was searched with Mash Screen [15] in the PLSDB database (a resource containing 16 168 plasmid sequences collected from the NCBI nucleotide database) [16]. Four additional IncHI2 plasmids recovered from PLSDB were used for the BRIG comparison analysis: pSTm-A54650 (*S.* Typhimurium A54560) from Malawi, pEB247 (*Enterobacter bugandensis* EB-147) from the United Republic of Tanzania, pCRENT-193_1 (*Enterobacter* sp. CRENT-193) from the Republic of Korea and pEc21617-310 (*Escherichia coli* Ec21617) from Taiwan (Table 1).

The GenBank accession numbers of all the genomes studied are listed in Table 1.

## Statistical analysis

The chi-square test for trend was used to compare rates of resistance to antibiotics during the study period. P values < 0.05 were considered statistically significant. All analyses were performed with STATA 12.0 (Stata Corporation, College Station, TX, USA).

## Results

### Serotype distribution and antimicrobial susceptibility

A total of 582 non-duplicate isolates of *Salmonella* belonging to 113 serotypes were collected during the 10-year study period. The most common serotypes were Typhimurium (n = 180, 31% of the isolates), Enteritidis (n = 64, 11%) and Typhi (n = 35, 6%) (Table 2). No strains assigned to serotypes Paratyphi A and Paratyphi B were detected. Typhimurium was the most frequently recovered serotype each year, except in 2010 (rank 2) and in 2013 (not found). Isolates were from stool (n = 452, 77%), blood (n = 97, 17%), urine (n = 14, 2%), cerebrospinal fluid (n = 13, 2%), wound (n = 5, 1%), and pleural fluid (n = 1) samples. Of the blood isolates, 77 (79%) were NTS, and Typhimurium (n = 49, 64%% of the isolates) and Enteritidis (n = 15, 19%) were the predominant serotypes. The other NTS serotypes were Dublin (n = 6), Stanleyville (n = 3), Brancaster (n = 1), Hartford (n = 1), Mikawashima (n = 1) and Saintpaul (n = 1). Three blood isolates were not typables and 17 belonged to serotype Typhi. A total of 13 cases of meningitis due to *Salmonella enterica* were reported during the study period, and Typhimurium was involved in 9 cases (69%). The other serotypes were Enteritidis (n = 1), Poona (n = 1), Sainpaul (n = 1) and Typhi (n = 1).

Overall, the 582 *S. enterica* isolates displayed a low level of resistance to all antibiotics, except those used as traditional first-line drugs: tetracycline (18%), chloramphenicol (30%), cotrimoxazole (39%) and amoxicillin (40%) (Table 3). The most frequent phenotype of resistance in serotypes Enteritidis (22/64, 34%) and Typhi (9/38, 24%) was susceptibility to all the beta-lactams tested, except for amoxicillin and ticarcillin, to aminoglycosides, to fluoroquinolones, and resistance to chloramphenicol, cotrimoxazole and tetracyclin. The susceptibility of *S.* Typhi to antibiotics was not significantly different from that of NTS (P > 0.5).

One stool/blood isolate with serotype Kentucky collected in 2010 was resistant to ciprofloxacin (MICs: > 12 mg/L, respectively). The blood Typhimurium isolate S1027072 was resistant to the beta-lactams tested (ceftazidime: MIC 32 mg/L, ceftriaxone: MIC >256 mg/L), except for cefoxitin and imipenem, to gentamicin, to chloramphenicol, cotrimoxazole, and

**Table 2. Distribution of the 10 most frequent *Salmonella enterica* serotypes in Central African Republic, 2004–2013.**

| 2004 n=46 | % | 2005 n=81 | % | 2006 n=100 | % | 2007 n=97 | % | 2008 n=37 | % | 2009 n=61 | % | 2010 n=39 | % | 2011 n=63 | % | 2012 n=43 | % | 2013 n=15 | % | 2004–2013 n=582 | % |
|---|---|---|---|---|---|---|---|---|---|---|---|---|---|---|---|---|---|---|---|---|---|
| Typhimurium | 28 | Typhimurium | 40 | Typhimurium | 40 | Typhimurium | 29 | Typhimurium | 50 | Typhimurium | 46 | Enteritidis | 31 | Typhimurium | 18 | Typhimurium | 10 | Liverpool | 27 | Typhimurium | 31 |
| Typhi | 13 | Enteritidis | 12 | Enteritidis | 12 | Enteritidis | 16 | Typhi | 15 | Enteritidis | 8 | Typhimurium | 11 | Onireke | 13 | Nchanga | 8 | Kibusi | 13 | Enteritidis | 11 |
| Enteritidis | 11 | Typhi | 6 | Typhi | 6 | Typhi | 7 | 42:r:-[a] | 13 | Stanleyville | 5 | Typhimurium | 10 | Enteritidis | 8 | Mikawasima | 6 | Hartford | 7 | Typhi | 6 |
| Mgulani | 4 | Saintpaul | 6 | Stanleyville | 6 | Dublin | 5 | Mikawasima | 5 | Kibusi | 5 | Kibusi | 5 | Stanleyville | 8 | Stanleyville | 6 | Texas | 7 | Stanleyville | 3 |
| 42:r:-[a] | 2 | Stanleyville | 5 | 42:r:-[a] | 5 | Stanleyville | 5 | Amoundernes | 1 | Llandoff | 5 | 42:r:-[a] | 5 | Give | 5 | Onireke | 6 | Hvittingfoss | 7 | 42:r:-[a] | 2 |
| Ayton | 2 | Dublin | 4 | Infantis | 4 | 42:r:-[a] | 4 | Stanleyville | 1 | Infantis | 5 | Hull | 3 | Chicago | 5 | Liverpool | 5 |  |  | Kibusi | 2 |
| Bovismorbificans | 2 | Hartford | 2 | Poona | 2 | Schwarzengrund | 4 | Enteritidis | 1 | Typhi | 3 | Saintpaul | 2 | Stanleyville | 5 | Sinstorf | 3 |  |  | Onireke | 2 |
| Budapest | 2 | 42:r:-[a] | 2 | Hull | 2 | Leeuwarden | 4 | Schwarzengrund | 1 | 42:r:-[a] | 3 | Typhi | 2 | Mikawasima | 3 | Kentucky | 3 |  |  | Saintpaul | 2 |
| Butantan | 2 | Leeuwarden | 2 | Saintpaul | 1 | Mikawasima | 3 | Mgulani | 1 | Amoundernes | 3 | Hadar | 2 | Kibusi | 3 | Llandoff | 3 |  |  | Stanleyville | 2 |
| Chile | 2 | Infantis | 2 | Hadar | 1 | Stanley | 2 | Aberdeen | 1 | Poona | 3 | Onireke | 2 | Ilugun | 3 | Kingston | 3 |  |  | Mikawasima | 1 |

[a] subsp. *salamae*

**Table 3. Percentage resistance to specific antibiotics in *Salmonella enterica* in the Central African Republic, 2004–2013.**

| Antibiotic | 2004 | 2005 | 2006 | 2007 | 2008 | 2009 | 2010 | 2011 | 2012 | 2013 | Total |
|---|---|---|---|---|---|---|---|---|---|---|---|
| | *n* = 46 | *n* = 81 | *n* = 100 | *n* = 97 | *n* = 37 | *n* = 61 | *n* = 39 | *n* = 63 | *n* = 43 | *n* = 15 | *n* = 582 |
| Amoxicillin | 46 | 51 | 41 | 62 | 51 | 26 | 5 | 13 | 44 | 29 | 40 |
| Amoxicillin-clavulanic acid | 4 | 1 | 3 | 0 | 3 | 2 | 0 | 0 | 2 | 0 | 2 |
| Ticarcillin | 46 | 51 | 41 | 62 | 51 | 26 | 5 | 13 | 44 | 29 | 40 |
| Cefalotin | 0 | 1 | 0 | 0 | 3 | 3 | 0 | 0 | 2 | 0 | 1 |
| Cefoxitin | 0 | 0 | 0 | 0 | 0 | 0 | 0 | 0 | 2 | 0 | 0 |
| Cefotaxime | 0 | 0 | 0 | 0 | 3 | 0 | 0 | 0 | 0 | 0 | 0 |
| Imipenem | 0 | 0 | 0 | 0 | 0 | 0 | 0 | 0 | 0 | 0 | 0 |
| Amikacin | 0 | 0 | 0 | 0 | 0 | 0 | 0 | 0 | 0 | 0 | 0 |
| Gentamicin | 2 | 0 | 2 | 0 | 3 | 2 | 5 | 2 | 0 | 0 | 1 |
| Nalidixic acid | 0 | 0 | 3 | 0 | 8 | 2 | 8 | 2 | 0 | 0 | 2 |
| Ciprofloxacin | 0 | 0 | 0 | 0 | 3 | 0 | 3 | 0 | 0 | 0 | 0 |
| Chloramphenicol | 28 | 33 | 32 | 54 | 43 | 25 | 26 | 19 | 40 | 20 | 30 |
| Cotrimoxazole | 41 | 52 | 44 | 61 | 51 | 25 | 3 | 11 | 44 | 27 | 39 |
| Tetracycline | 20 | 22 | 25 | 28 | 22 | 13 | 5 | 6 | 9 | 7 | 18 |

tetracyclin, and susceptible to amikacin, nalidixic acid, and ciprofloxacin. The DDS test was positive, indicating production of an ESBL.

## Genetic background of *Salmonella* Typhimurium and antibiotic susceptibility

CRISPOL typing was performed on a set of 68 *S.* Typhimurium isolates, including S1027072. They were chosen using a random number table and corresponded to around 38% of the isolates collected every year. They were from stools (n = 39, 57%), blood (n = 17, 25%), cerebrospinal fluid (n = 5, 7%), urine (n = 3, 4%), wounds (n = 3, 4%) and pleural fluid (n = 1). *S.* Typhimurium was isolated from both stools and blood in 13 cases.

A total of 66 isolates (96%) belonged to CRISPOL type (CT) 28 group, of which 63 (97%) were assigned to CT28, two to CT698 and one to CT526. It was shown previously that CT28 is associated with lineage II, whereas CT698 is associated with lineage I [17] (Table 4). To confirm the association of ST313 to the CT28 group, MLST was performed on 10 isolates: 7 assigned to CT28, one to CT698 and one to CT526. All the isolates belonged to ST313. In addition, the ESBL-*S.* Typhimurium belonging to CT28 by CRISPOL typing was assigned to ST313 by WGS. All these data confirmed the association between CT28 group and ST313 in *Salmonella* isolates from CAR.

Additional antibiotics susceptibility tests were performed to streptomycin and spectinomycin. The most common resistance phenotype was to amoxicillin, ticarcillin, cotrimoxazole, chloramphenicol and streptomycin (35/66, 53%). All but one of the CT28 group isolates were resistant to at least three antibiotics (Table 4).

All the invasive *S.* Typhimurium isolates except one (CT301) were assigned to ST313 (CT28).

## Plasmid carriage in the ESBL-producing *S.* Typhimurium isolate S1027072

Plasmid Finder-based typing showed the presence of plasmids of three incompatibility groups, IncHI2, IncFIB and IncQ1. The IncHI2 plasmid was assigned to ST1. Use of a mapping

**Table 4. Characteristics of 68 randomly chosen *Salmonella* Typhimurium isolates in Bangui, 2004–2013.**

| Resistance type | Associated with sequence type (ST) 313 | | | | |
| --- | --- | --- | --- | --- | --- |
| | Yes | | | No | |
| | CRISPOL type (CT) 28[a] | CT526[a] | CT698[b] | CT96 | CT301 |
| ATC | 1 | | | | |
| ATSNalSXTC | 1 | | | | |
| ATSpSXTCTe | 1 | | | | |
| ATSSpKTSXT | | | | | |
| ATSSpSXT | | | 1 | | |
| ATSSpSXTC | 1 | | 1 | | |
| ATSSpSXTCTe | 2 | | | | |
| ATSSXT | 3 | | | | |
| ATSSXTC | 35 | 1 | | | |
| ATSSXTCTe | 11 | | | | |
| ATSSXTTe | 3 | | | | |
| ATSXTCTe | 1 | | | | |
| ATSXTTe | | | | 1 | |
| SSpSXT | 1 | | | | |
| SSXT | 2 | | | | |
| SSXTCTe | 1 | | | | |
| Susceptible | | | | | 1 |
| Total | 63 | 1 | 2 | 1 | 1 |

A, amoxicillin; C, chloramphenicol; Nal, Nalidixic acid; S, streptomycin; Sp, spectinomycin; SXT, cotrimoxazole; Te, tetracycline, T, ticarcillin.

[a]Associated with ST313 lineage II

[b]Associated with ST313 lineage I

approach between plasmids of the ESBL-producing *S.* Typhimurium published genome KST313 (pKST313 and pSBLT) and isolate S1027072 made it possible to form two putative plasmidic assemblies, pCARST313 (IncHI2) and pCAR_SBLT (IncF).

The first plasmidic assembly pCARST313 presents 99.9% nucleotidic identity with pKST313 (300 375 bp). Both plasmids encode: (i) resistance to heavy-metal ions, including mercury (*mer* and *tni* genes), tellurite (*ter* genes), arsenic (*ars* genes), copper (*cusS* and *pcoE* genes), nickel and cobalt (*rcnR* and *rcnA* genes), and to tellurite utilization genes (*tel* genes); and (ii) resistance to antimicrobials, including aminoglycosides (*aac*(6')-Ib), streptomycin (*aadA1*, *strA* and *strB*), beta-lactams ($bla_{OXA-30}$, $bla_{TEM-1}$ and $bla_{CTX-M-15}$), chloramphenicol (*catA1* and *catB3*), trimethoprim (*dfrA14*), sulfonamides (*sulI* and *sulII*), gentamicin (*aacC3*) and tunicamycin (*tmrB*), most of which are in a class 1 integron. Genes for conjugative transfer were also identified. The transferability of the plasmid was confirmed in the conjugation experiment. We did not find any spontaneous chromosomal mutations associated with resistance to antibiotics. To explore putative circulation of close $bla_{CTX-M-15}$-carrying IncHI2 plasmids identical to pCARST313 in sub-Saharan Africa, we assembled this plasmid from the strains *S.* Havana 07–0319 and *S.* Telelkebir 07–1331 isolated in Mali in 2007, *S.* Grumpensis 08–3663 collected in Senegal in 2008 and *S.* Typhimurium A54560 collected in Malawi in 2009. The plasmid was recovered from all isolates with a minimal identity of 99% (Table 1) and was also found among Enterobacteriaceae species: *Enterobacter bugandensis* collected in the United Republic of Tanzania in 2010, *Enterobacter* sp. isolated in the Republic of Korea in 2013 and *Escherichia coli* isolated in Taiwan in 2013 (Fig 1).

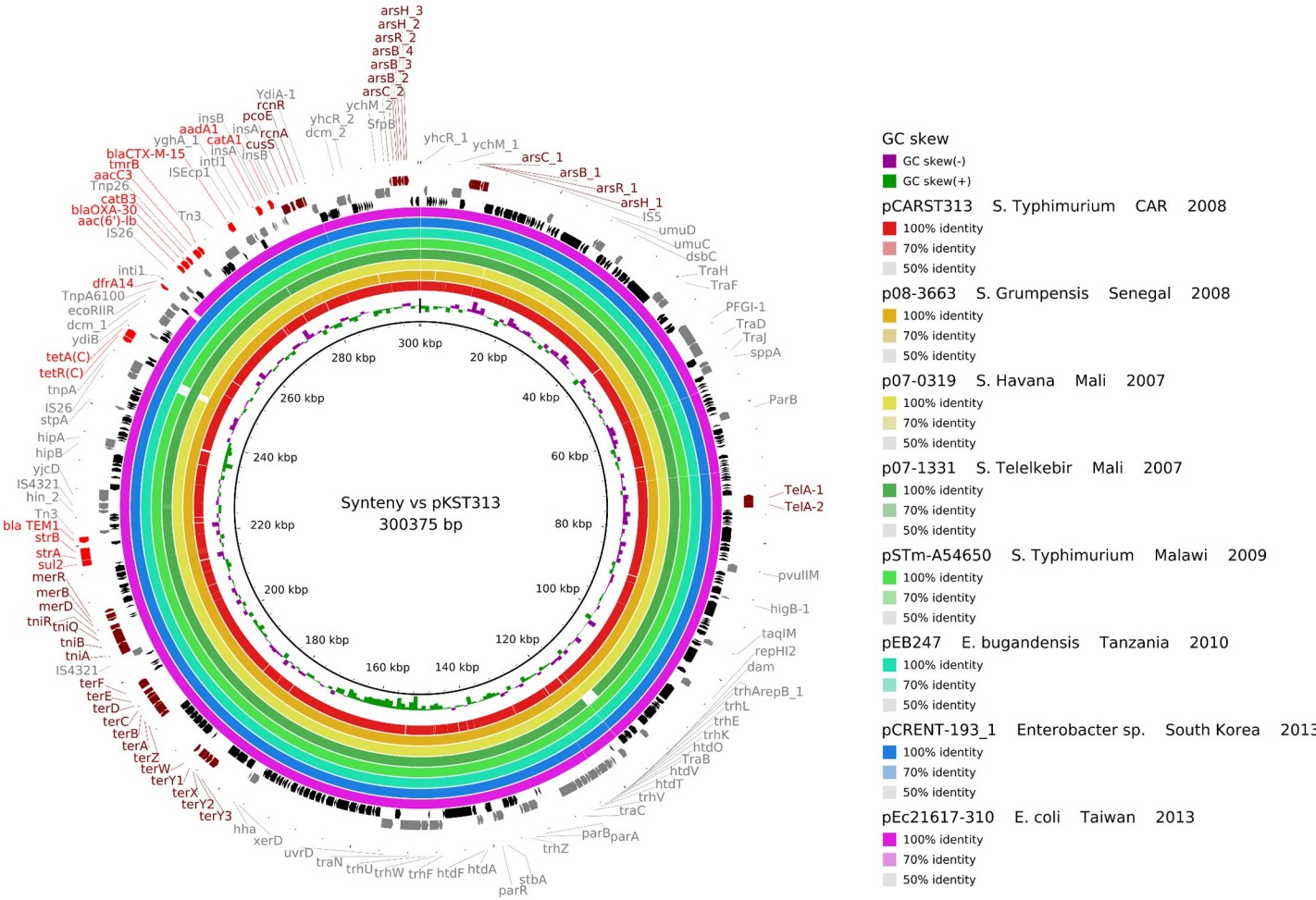

**Fig 1. Syntenic analysis of pKST313 vs 8 other IncHI2 plasmids.** The innermost black ring 1 represents the reference sequence of pKST313. Following rings correspond to pairwise comparison with IncHI2 plasmids: ring 2 represents pCARST313; ring 3, p08-3663; ring 4, p07-0319; ring 5, p07-1331; ring 6, pSTm-A54650; ring 7, pEB247; ring 8, pCRENT-193_1; ring 9, pEc21617-310. Last 4 rings represent genetic map of pKST313: antibiotic resistance genes are indicated by red arrows, heavy-metals resistance genes by brown arrows, others annotated genes by grey arrows and hypothetical proteins by black arrows.

The second plasmidic assembly pCAR_SBLT, found in the S1027072 strain, also presented > 99% nucleotide identity with pSBLT (128 925bp) found in ST313. Both plasmids encode genes involved in antibiotic resistance: *catA1*, *sulI*, *sulII*, *bla*$_{TEM-1}$, *strA*, *strB*, *dhfrI*, *aadA1* and *qacED1*. This plasmid was not found in other *Salmonella* isolates harbouring the pKST313 plasmid type.

## Discussion

In this study of 582 non-duplicate *S. enterica* isolates investigated between 2004 and 2013, Typhimurium and Enteritidis were the two most frequently recovered NTS serotypes, in accordance with numerous other studies in sub-Saharan Africa [1–3, 18–23]. Surprisingly, they were not found in 2013, probably due to the small number of *Salmonella* isolates collected this year. Indeed, significant differences in the numbers of organisms collected were observed by year, due to the economic and political crisis during the study period, which further restricted patient access to health care facilities. Since 2013, the CAR has faced an ongoing civil war, plunging the country into a chaotic state of violence and an ensuing humanitarian crisis.

The surveillance of *Salmonella enterica* isolates at the Institut Pasteur in Bangui was discontinued but a WGS approach-study could be coordinated when the laboratory-surveillance capacities will be strengthened locally.

Overall, the resistance rates to all antibiotic classes were low, except to the antibiotics used previously (amoxicillin, chloramphenicol, cotrimoxazole and tetracycline), confirming recommendations for first-line treatment based on C3G and fluoroquinolones. We report the first case of ESBL-producing *Salmonella* (serotype Typhimurium) in CAR, which is of concern. More new descriptions of ESBL-producing *Salmonella* isolates have been made across sub-Saharan Africa since the first, in 2006 in Ethiopia, Typhimurium being the most frequently isolated ESBL-producing serotype [3, 24–26]. The low prevalence (0–1.3%) described in sub-Saharan Africa may not, however, represent the situation at country level, as most countries do not have national surveillance systems based on networks of clinical laboratories.

*S.* Typhimurium ST313 was the primary cause of invasive salmonellosis during the study period, in agreement with data from Congo (Central Africa), Kenya, Malawi and Mozambique (East Africa) [17, 23, 27]. The high proportion of *S.* Typhimurium ST313 among stools in our study suggests that this clone causes also an appreciable burden of gastroenteritis. A human reservoir for this clone is plausible, especially as identical strains were recovered in asymptomatic siblings and parents of index case-patients in Kenya (carriage prevalence, 6.9%) [28]. Our data also confirm the substantial prevalence of multi-drug-resistant *S.* Typhimurium ST313 [22, 27]. This is of major concern, and further investigations are required to identify the reservoirs and risk factors for exposure and transmission in order to design hypothesis-driven preventive measures.

The invasive ESBL-producing *S.* Typhimurium strain S1027072 was assigned to the highly invasive ST313 genetic background. The $bla_{CTX-M-15}$ gene and most antimicrobial resistance genes were present in a large conjugative IncHI2 plasmid, which was closely similar to the pKST313 plasmid found in an ESBL-producing *S.* Typhimurium ST313 isolate in Kenya [9] and also to ESBL-carrying plasmids found in three other *Salmonella* serotypes collected in Senegal, Mali and Malawi, highlighting its significant role in the spread of $bla_{CTX-M-15}$ genes in *Salmonella* isolates in sub-Saharan Africa. Its significance may be even greater, as it has been recovered in various *Enterobacteriaceae* species in the United Republic of Tanzania and east Asia. In addition, this plasmid has numerous genes encoding resistance to heavy-metal ions. It is well known that the location of both metals and antibiotic resistance genes on the same mobile element play a major role in the persistence, selection and spread of antibiotic-resistant bacteria in anthropogenic environments heavily contaminated with detergents, heavy metals and other antimicrobials [29, 30]. In developing countries, rivers, lakes and lagoons are often contaminated with untreated hospital and industrial effluents and also by urban storm-water containing anthropogenic pollutants due to intensive uncontrolled urbanization. This is optimal conditions for bacterial development and the spread of antibiotic-resistant bacteria. Further active surveillance is needed to minimize the spread of this successful IncHI2 plasmid in order to control dissemination of antibiotic resistance among *Salmonella* isolates. Unsurprisingly, a plasmid similar to the pSLT-BT virulence plasmid usually found in *S.* Typhimurium ST313 lineage II harbouring integron-borne resistance determinants was found in the ESBL-producing *S.* Typhimurium isolate S1027072 in CAR [9].

Our results confirm high rates of invasive multidrug-resistant *S.* Typhimurium ST313 among *Salmonella* infections in CAR between 2004 and 2013. Although the prevalence of ESBL-producing *Salmonella enterica* isolates was very low, the wide distribution of a single IncHI2 plasmid scaffold among *Salmonella* serotypes in sub-Saharan Africa, the origin of dissemination of the $bla_{CTX-M-15}$ gene, is of concern.

## Acknowledgments

We thank all the technicians at the Institut Pasteur *Salmonella* laboratories in Bangui and in Paris for serotyping the isolates (preliminary and complete serotyping, respectively); L. Fabre and V. Enouf from Institut Pasteur in Paris for CRISPOL typing and whole genome sequencing of the isolates, respectively. We also thank E. Heseltine for editorial assistance.

## Author Contributions

**Conceptualization:** Sebastien Breurec, Simon Le Hello.

**Data curation:** Sebastien Breurec, Yann Reynaud, François-Xavier Weill.

**Formal analysis:** Sebastien Breurec, Yann Reynaud, Geoffrey Costilhes, François-Xavier Weill, Simon Le Hello.

**Funding acquisition:** François-Xavier Weill, Simon Le Hello.

**Investigation:** Sebastien Breurec, Yann Reynaud, Thierry Frank, Alain Farra, Geoffrey Costilhes, François-Xavier Weill, Simon Le Hello.

**Methodology:** Sebastien Breurec, Yann Reynaud, Geoffrey Costilhes, François-Xavier Weill, Simon Le Hello.

**Project administration:** Simon Le Hello.

**Resources:** Thierry Frank, Alain Farra, François-Xavier Weill, Simon Le Hello.

**Software:** Yann Reynaud.

**Supervision:** Sebastien Breurec, Alain Farra, François-Xavier Weill.

**Validation:** Sebastien Breurec, François-Xavier Weill, Simon Le Hello.

**Writing – original draft:** Sebastien Breurec, Yann Reynaud, François-Xavier Weill, Simon Le Hello.

**Writing – review & editing:** Thierry Frank, Alain Farra, Geoffrey Costilhes.

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
