## [Decision Letter · Decision Letter 0]

20 Aug 2019

Dear Dr Breurec:

Thank you very much for submitting your manuscript "10-year surveillance of Salmonella enterica in Bangui, Central African Republic, 2004–2013" (PNTD-D-19-00975) for review by PLOS Neglected Tropical Diseases. Your manuscript was fully evaluated at the editorial level and by independent peer reviewers. The reviewers appreciated the attention to an important topic but identified some aspects of the manuscript that should be improved.

We therefore ask you to modify the manuscript according to the review recommendations before we can consider your manuscript for acceptance. Your revisions should address the specific points made by each reviewer.

(1) A letter containing a detailed list of your responses to the review comments and a description of the changes you have made in the manuscript.

(2) Two versions of the manuscript: one with either highlights or tracked changes denoting where the text has been changed (uploaded as a "Revised Article with Changes Highlighted" file ); the other a clean version (uploaded as the article file).

(3) If available, a striking still image (a new image if one is available or an existing one from within your manuscript). If your manuscript is accepted for publication, this image may be featured on our website. Images should ideally be high resolution, eye-catching, single panel images; where one is available, please use 'add file' at the time of resubmission and select 'striking image' as the file type. 

Please provide a short caption, including credits, uploaded as a separate "Other" file. If your image is from someone other than yourself, please ensure that the artist has read and agreed to the terms and conditions of the Creative Commons Attribution License at http://journals.plos.org/plosntds/s/content-license (NOTE: we cannot publish copyrighted images). 

(4) Appropriate Figure Files 

Please remove all name and figure # text from your figure files upon submitting your revision. Please also take this time to check that your figures are of high resolution, which will improve both the editorial review process and help expedite your manuscript's publication should it be accepted. Please note that figures must have been originally created at 300dpi or higher. Do not manually increase the resolution of your files. For instructions on how to properly obtain high quality images, please review our Figure Guidelines, with examples at: http://journals.plos.org/plosntds/s/figures

While revising your submission, please upload your figure files to the Preflight Analysis and Conversion Engine (PACE) digital diagnostic tool, https://pacev2.apexcovantage.com/ PACE helps ensure that figures meet PLOS requirements. To use PACE, you must first register as a user. Then, login and navigate to the UPLOAD tab, where you will find detailed instructions on how to use the tool. If you encounter any issues or have any questions when using PACE, please email us at figures@plos.org.

We hope to receive your revised manuscript by Oct 19 2019 11:59PM. If you anticipate any delay in its return, we ask that you let us know the expected resubmission date by replying to this email.

To submit your revised files, please log in to https://www.editorialmanager.com/pntd/

Sincerely,

Elsio Wunder Jr, D.V.M., Ph.D.

Deputy Editor

Elsio Wunder Jr

Deputy Editor

Reviewer's Responses to Questions

**Key Review Criteria Required for Acceptance?**

**Methods**

-Are the objectives of the study clearly articulated with a clear testable hypothesis stated?

-Is the study design appropriate to address the stated objectives?

-Is the population clearly described and appropriate for the hypothesis being tested?

-Is the sample size sufficient to ensure adequate power to address the hypothesis being tested?

-Were correct statistical analysis used to support conclusions?

-Are there concerns about ethical or regulatory requirements being met?

Reviewer #1: This study, including a large number of strains describe the epidemiology of Salmonella infections in CAR between 2004 and 2013. His strength is to be interested in strain genetics. Furthermore, data concerning the antimicrobial resistance in this part of the world are substancial.

The manuscript is clear and easy to read and objectives are clearly defined.

Reviewer #2: Line 99: Please clarify what is meant by "no additional specimens were collected for this study". At this status of the manuscript it is not clear/known to the reader which samples actually were collected.

L101: To which healthcare facility are authors referring to? Which type of patients (including their eligibility criteria) and infectious/chronic diseases are under the routine surveillance mentioned?

L111: From which biological samples were the isolates detected?

L114-117: Which antibiotic concentration were tested?

**Results**

-Does the analysis presented match the analysis plan?

-Are the results clearly and completely presented?

-Are the figures (Tables, Images) of sufficient quality for clarity?

Reviewer #1: The analysis are in accord with the plan and the results are clearly presented and easy to read. 

- The Table 2 could be enlarged

- It would be interesting to detail the various serotypes of Salmonella responsible for bacteremia and meningitis

- Can the authors specify in which site (stool, blood or other ) the Salmonella producing ESBL was isolated?

Reviewer #2: L 183-184: So, 9/13 patients with meningitis had a co-infection with S. Typhimurium?

L 193: Please clarify what is meant by low resistance. Were the isolates tested resistant to the antibiotics tested or did they possibly show some borderline/intermediate patterns? Which standards were used as a reference to classify isolates based on their susceptibility/resistant patterns?

**Conclusions**

-Are the conclusions supported by the data presented?

-Are the limitations of analysis clearly described?

-Do the authors discuss how these data can be helpful to advance our understanding of the topic under study?

-Is public health relevance addressed?

Reviewer #1: The conclusions are supported by the data. These data concerning the epidemiology of Salmonella nfections in the the sub-Saharan Africa are significant. Antimicrobial resistance data are very important for first-line treatments.

Reviewer #2: NA

**Editorial and Data Presentation Modifications?**

Reviewer #1: This study is well designed and well conducted and the manuscript is clear. 

It can be accepted

Reviewer #2: NA

**Summary and General Comments**

Reviewer #1: (No Response)

Reviewer #2: Congratulations to this well designed and written article. I have just some very few suggestions related to the methods and results sections.

PLOS authors have the option to publish the peer review history of their article (what does this mean?). If published, this will include your full peer review and any attached files.

Reviewer #1: Yes: Dr Josette Raymond

Reviewer #2: No

---

## [Decision Letter · Decision Letter 1]

26 Sep 2019

Dear Dr Breurec:

Thank you very much for submitting your manuscript "10-year surveillance of Salmonella enterica in Bangui, Central African Republic, 2004–2013" (PNTD-D-19-00975R1) for review by PLOS Neglected Tropical Diseases. Your manuscript was fully evaluated at the editorial level and by independent peer reviewers. The reviewers appreciated the attention to an important topic but identified some aspects of the manuscript that should be improved.

We therefore ask you to modify the manuscript according to the review recommendations before we can consider your manuscript for acceptance. Your revisions should address the specific points made by each reviewer.

(1) A letter containing a detailed list of your responses to the review comments and a description of the changes you have made in the manuscript.

(2) Two versions of the manuscript: one with either highlights or tracked changes denoting where the text has been changed (uploaded as a "Revised Article with Changes Highlighted" file ); the other a clean version (uploaded as the article file).

(3) If available, a striking still image (a new image if one is available or an existing one from within your manuscript). If your manuscript is accepted for publication, this image may be featured on our website. Images should ideally be high resolution, eye-catching, single panel images; where one is available, please use 'add file' at the time of resubmission and select 'striking image' as the file type. 

Please provide a short caption, including credits, uploaded as a separate "Other" file. If your image is from someone other than yourself, please ensure that the artist has read and agreed to the terms and conditions of the Creative Commons Attribution License at http://journals.plos.org/plosntds/s/content-license (NOTE: we cannot publish copyrighted images). 

(4) Appropriate Figure Files 

Please remove all name and figure # text from your figure files upon submitting your revision. Please also take this time to check that your figures are of high resolution, which will improve both the editorial review process and help expedite your manuscript's publication should it be accepted. Please note that figures must have been originally created at 300dpi or higher. Do not manually increase the resolution of your files. For instructions on how to properly obtain high quality images, please review our Figure Guidelines, with examples at: http://journals.plos.org/plosntds/s/figures

While revising your submission, please upload your figure files to the Preflight Analysis and Conversion Engine (PACE) digital diagnostic tool, https://pacev2.apexcovantage.com/ PACE helps ensure that figures meet PLOS requirements. To use PACE, you must first register as a user. Then, login and navigate to the UPLOAD tab, where you will find detailed instructions on how to use the tool. If you encounter any issues or have any questions when using PACE, please email us at figures@plos.org.

We hope to receive your revised manuscript by Nov 25 2019 11:59PM. If you anticipate any delay in its return, we ask that you let us know the expected resubmission date by replying to this email.

To submit your revised files, please log in to https://www.editorialmanager.com/pntd/

Sincerely,

Florian Marks, Ph.D., MPH

Guest Editor

Elsio Wunder Jr

Deputy Editor

Reviewer's Responses to Questions

**Key Review Criteria Required for Acceptance?**

**Methods**

-Are the objectives of the study clearly articulated with a clear testable hypothesis stated?

-Is the study design appropriate to address the stated objectives?

-Is the population clearly described and appropriate for the hypothesis being tested?

-Is the sample size sufficient to ensure adequate power to address the hypothesis being tested?

-Were correct statistical analysis used to support conclusions?

-Are there concerns about ethical or regulatory requirements being met?

Reviewer #1: The manuscript was corrected according to the reviewer's comments and can now be accepted

Reviewer #2: No further comments.

Reviewer #3: This a descriptive microbiological analysis of routine Salmonella isolates from an admittedly very understudied setting in the CAR with then some more detailed information on a single ESBL isolate. While the study thus provides some new data on Salmonella in this geographic area, it also suffers from a number of major issues

1) There is no clear testable hypothesis stated. The provides a descriptive analysis of routinely collected isolates from a single clinical microbiology lab.

2) A description of how the samples came into the Pasteur lab is missing. Is the Institut Pasteur lab associated with a health clinic, hospital, private GPs? Did some isolates derive from the mentioned case-control study conducted in 2011-13? If yes, how many isolates from each "source"? What type of cases (invasive, stool) are these isolates most likely to be representative of?

3) Given that the title mentions "surveillance", one would at least expect a summary epidemiological analysis including age distribution & sex of cases, in addition to providing microbiological data like serotyping and antibiotic resistance testing on non-duplicate isolates. Please revise title to include "serotyping", "antimicrobial resistance" and "human" isolates 

4) The data collection period and the typing technique CRISPOL used are starting to be dated. Would it not be possible to extend the analysis to isolates also from the last 5 years, 2014-2018 and also provide more WGS results if possible. The authors need to say at least whether Salmonella "surveillance" is still ongoing now or was discontinued after 2013.

5) The sample size of isolates from 2013 in table 2 is rather small compared to other years (and the serotype distribution is quite different to previous years - was S Liverpool really the top serovar?). Reasons for this needs to be at least discussed in the manuscript.

6) randomization procedure for selecting S. Typhimurium isolates needs to be explained more. 

7) It is not clear where the various lab testing done, was all serotyping and AMR testing done in Paris at the end of the study period ?

8) No description was given of how the MLST type (ST313) was obtained. I guess that the authors assumed that the CT28 is indicative of ST313, rather than having conducted MLST, i.e the ST was "deduced" based on results from a previous study in the Democratic Republic in Congo. This needs to be made clearer in methods. I recommend that authors change wording in abstract, replacing "associated with" by "indicative of". One of the problems with this approach is that the correspondence between CT28 and ST313 might be local to the Congo area. Ideally at least some CT28 S. Typhimurium isolates should be sequenced by WGS to make sure that they also correspond to ST313 in the CAR. For the ESBL isolate submitted to WGS, the authors should also individually report for completeness the full antibiotic resistance profile, the ST, as well as any non-ESBL resistance genes or mutations, for example using ResFinder or CARD.

**Results**

-Does the analysis presented match the analysis plan?

-Are the results clearly and completely presented?

-Are the figures (Tables, Images) of sufficient quality for clarity?

Reviewer #1: The manuscript was corrected according to the reviewer's comments and can now be accepted

Reviewer #2: No further comments.

Reviewer #3: In table 1, it's not clear why the size information of the plasmid is not given for strains in lines 2 to 5 - is it because it is not a full assembly? The authors could report the size of the de novo assembly and then indicate as a footnote, that it might not be complete.

In most developed countries, resistance patterns are strongly associated with serotype, typically higher for S. Typhimurium and Typhi than for other serovars. The authors do not provide any detailed AMR data for serovars other than S. typhimurium, like S. Enteritidis and S. Typhi which are also quite common. A new table should be added with showing most frequent AMR profiles for most frequent serotypes, or at least mention most most frequent AMR profiles for most frequent serovars in the text.

In table 4, for the footnotes a & b, need to add citation to paper showing the association with ST313

**Conclusions**

-Are the conclusions supported by the data presented?

-Are the limitations of analysis clearly described?

-Do the authors discuss how these data can be helpful to advance our understanding of the topic under study?

-Is public health relevance addressed?

Reviewer #1: The manuscript was corrected according to the reviewer's comments and can now be accepted

Reviewer #2: No further comments.

Reviewer #3: The authors interpretation that one isolate of 582 (0.2%) being an ESBL is a "major" concern is debatable. One could also sonclude that ESBL in Salmonella in this setting are quite rare, and probably rarer than other Enterobacteriacea in other African countries. Consider replacing "major concern" by "concern".

Line 295 the authors write that clinical features are unknown and they say in the same sentence that they reported diarrhea. This doesn't make sense. Either they have (some of) this data or they don't.

Line 300 the authors mention the "emergence" of MDR ST313 which suggests a change in frequency over time. The data presented herein does not support such a changing trend over time, so authors should consider replacing "emergence" with "substantial prevalence of". Also the last sentence about more frequent" is quite speculative as the data presented in this study does not show any increase over time.

**Editorial and Data Presentation Modifications?**

Reviewer #1: The manuscript was corrected according to the reviewer's comments and can now be accepted

Reviewer #2: No further comments.

Reviewer #3: Acknowledgments: it's not usual to thank technicians for doing what appears to be "routine" work.

**Summary and General Comments**

Reviewer #1: It is a well conducted study, clearly written. It allows to precise the epidemiology of infections due to Salmonella in Central Africa Republic

Reviewer #2: All comments and suggestions raised during the 1st review were very well addressed. The manuscript reads very well.

Reviewer #3: (No Response)

PLOS authors have the option to publish the peer review history of their article (what does this mean?). If published, this will include your full peer review and any attached files.

Reviewer #1: No

Reviewer #2: No

Reviewer #3: No

---

## [Decision Letter · Decision Letter 2]

11 Nov 2019

Dear Dr Breurec,

We are pleased to inform you that your manuscript, "10-year surveillance of Salmonella enterica in Bangui, Central African Republic, 2004–2013", has been editorially accepted for publication at PLOS Neglected Tropical Diseases.

Before your manuscript can be formally accepted and sent to production you will need to complete our formatting changes, which you will receive in a follow up email. Please note: your manuscript will not be scheduled for publication until you have made the required changes.

IMPORTANT NOTES

* Copyediting and Author Proofs: To ensure prompt publication, your manuscript will NOT be subject to detailed copyediting and you will NOT receive a typeset proof for review. The corresponding author will have one final opportunity to correct any errors when sent the requests mentioned above. Please review this version of your manuscript for any errors.

* If you or your institution will be preparing press materials for this manuscript, please inform our press team in advance at plosntds@plos.org. If you need to know your paper's publication date for media purposes, you must coordinate with our press team, and your manuscript will remain under a strict press embargo until the publication date and time. PLOS NTDs may choose to issue a press release for your article. If there is anything that the journal should know, please get in touch.

*Now that your manuscript has been provisionally accepted, please log into EM and update your profile. Go to http://www.editorialmanager.com/pntd, log in, and click on the "Update My Information" link at the top of the page. Please update your user information to ensure an efficient production and billing process.

*Note to LaTeX users only - Our staff will ask you to upload a TEX file in addition to the PDF before the paper can be sent to typesetting, so please carefully review our Latex Guidelines [http://www.plosntds.org/static/latexGuidelines.action] in the meantime.

Best regards,

Florian Marks, Ph.D., MPH

Guest Editor

Elsio Wunder Jr

Deputy Editor

Reviewer's Responses to Questions

**Key Review Criteria Required for Acceptance?**

**Methods**

-Are the objectives of the study clearly articulated with a clear testable hypothesis stated?

-Is the study design appropriate to address the stated objectives?

-Is the population clearly described and appropriate for the hypothesis being tested?

-Is the sample size sufficient to ensure adequate power to address the hypothesis being tested?

-Were correct statistical analysis used to support conclusions?

-Are there concerns about ethical or regulatory requirements being met?

Reviewer #2: No further comments.

Reviewer #3: All points have been adequately addressed.

**Results**

-Does the analysis presented match the analysis plan?

-Are the results clearly and completely presented?

-Are the figures (Tables, Images) of sufficient quality for clarity?

Reviewer #2: No further comments.

Reviewer #3: All points have been adequately addressed.

**Conclusions**

-Are the conclusions supported by the data presented?

-Are the limitations of analysis clearly described?

-Do the authors discuss how these data can be helpful to advance our understanding of the topic under study?

-Is public health relevance addressed?

Reviewer #2: No further comments.

Reviewer #3: All points have been adequately addressed.

**Editorial and Data Presentation Modifications?**

Reviewer #2: No further comments.

Reviewer #3: All points have been adequately addressed.

**Summary and General Comments**

Reviewer #2: No further comments.

Reviewer #3: All points have been adequately addressed.

PLOS authors have the option to publish the peer review history of their article (what does this mean?). If published, this will include your full peer review and any attached files.

Reviewer #2: No

Reviewer #3: No

---

## [Editor Report · Acceptance letter]

26 Nov 2019

Dear Pr Breurec,

We are delighted to inform you that your manuscript, "Serotype distribution and antimicrobial resistance of human *Salmonella enterica* in Bangui, Central African Republic, from 2004 to 2013 ," has been formally accepted for publication in PLOS Neglected Tropical Diseases.

Best regards,

Serap Aksoy

Editor-in-Chief

Shaden Kamhawi

Editor-in-Chief
